# CogLTX: Applying BERT to Long Texts

**Ming Ding**
Tsinghua University
dm18@mails.tsinghua.edu.cn

**Chang Zhou**
Alibaba Group
ericzhou.zc@alibaba-inc.com

**Hongxia Yang**
Alibaba Group
yang.yhx@alibaba-inc.com

**Jie Tang**
Tsinghua University
jietang@tsinghua.edu.cn

## Abstract

BERT is incapable of processing long texts due to its quadratically increasing memory and time consumption. The most natural ways to address this problem, such as slicing the text by a sliding window or simplifying transformers, suffer from insufficient long-range attentions or need customized CUDA kernels. The maximum length limit in BERT reminds us the limited capacity ($5\sim 9$ chunks) of the *working memory* of humans —— then how do human beings **Cog**nize **L**ong **TeX**ts? Founded on the cognitive theory stemming from Baddeley [2], the proposed CogLTX [1] framework identifies key sentences by training a *judge* model, concatenates them for reasoning, and enables multi-step reasoning via *rehearsal* and *decay*. Since relevance annotations are usually unavailable, we propose to use interventions to create supervision. As a general algorithm, CogLTX outperforms or gets comparable results to SOTA models on various downstream tasks with memory overheads independent of the length of text.

## 1  Introduction

Pretrained language models, pioneered by BERT [12], have emerged as silver bullets for many NLP tasks, such as question answering [38] and text classification [22]. Researchers and engineers breezily build state-of-the-art applications following the standard finetuning paradigm, while might end up in disappointment to find some texts longer than the length limit of BERT (usually 512 tokens). This situation may be rare for normalized benchmarks, for example SQuAD [38] and GLUE [47], but very common for more complex tasks [53] or real-world textual data.

A straightforward solution for long texts is *sliding window* [50], processing continuous 512-token spans by BERT. This method sacrifices the possibility that the distant tokens "pay attention" to each other, which becomes the bottleneck for BERT to show its efficacy in complex tasks (for example Figure 1). Since the problem roots in the high $O(L^2)$ time and space complexity in transformers [46] ($L$ is the length of the

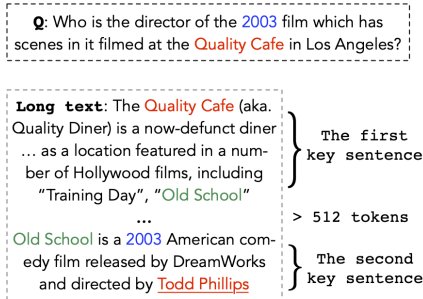

Figure 1: An example from HotpotQA (distractor setting, concatenated). The key sentences to answer the question are the first and last ones, more than 512 tokens away from each other. They never appear in the same BERT input window in the *sliding window* method, hence we fail to answer the question.

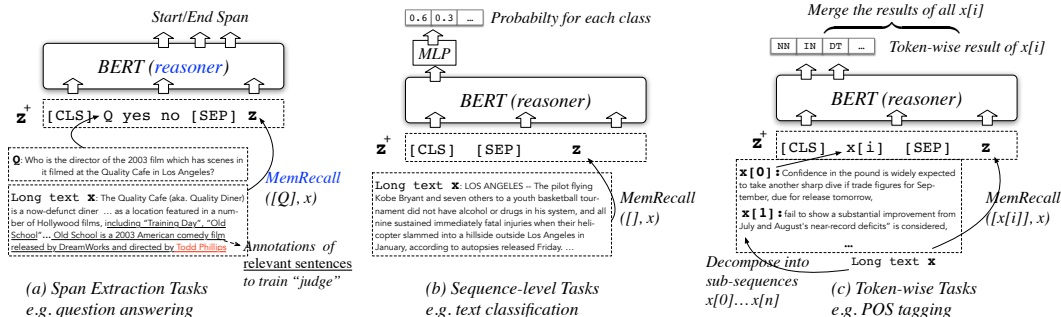

Figure 2: The CogLTX inference for main genres of BERT tasks. MemRecall is the process to extract key text blocks $\mathbf{z}$ from the long text $\mathbf{x}$. Then $\mathbf{z}$ is sent to the BERT, termed *reasoner*, to fulfill the specific task. A (c) task is converted to multiple (b) tasks. The BERT input w.r.t. $\mathbf{z}$ is denoted by $\mathbf{z}^{+}$.

text), another line of research attempts to simplify the structure of transformers [20, 37, 8, 42], but currently few of them have been successfully applied to BERT [35, 4].

The maximum length limit in BERT naturally reminds us the limited capacity of *Working Memory* [2], a human cognitive system storing information for logical reasoning and decision-making. Experiments [27, 9, 31] already showed that the working memory could only hold 5∼9 items/words during reading, so how do humans actually understand long texts?

"The central executive – the core of the (working memory) system that is responsible for coordinating (multi-modal) information", and "functions like a limited-capacity attentional system capable of selecting and operating control processes and strategies", as Baddeley [2] pointed out in his 1992 classic. Later research detailed that the contents in the working memory *decay* over time [5], unless are kept via *rehearsal* [3], i.e. paying attention to and refreshing the information in the mind. Then the overlooked information is constantly updated with relevant items from long-term memory by *retrieval competition* [52], collecting sufficient information for reasoning in the working memory.

The analogy between BERT and working memory inspires us with the CogLTX framework to **Cog**nize **L**ong **T**e**X**ts like human. The basic philosophy behind CogLTX is rather concise — reasoning over the concatenation of key sentences (Figure 2) — while compact designs are demanded to bridge the gap between the reasoning processes of machine and human.

The critical step in CogLTX is MemRecall, the process to identify relevant text blocks by treating the blocks as *episodic memories*. MemRecall imitates the working memory on retrieval competition, rehearsal and decay, facilitating multi-step reasoning. Another BERT, termed *judge*, is introduced to score the relevance of blocks and trained jointly with the original BERT *reasoner*. Moreover, CogLTX can transform task-oriented labels to relevance annotations by interventions to train *judge*.

Our experiments demonstrate that CogLTX outperforms or achieves comparable performance with the state-of-the-art results on four tasks, including NewsQA [44], HotpotQA [53], 20NewsGroups [22] and Alibaba, with constant memory consumption regardless of the length of text.

## 2 Background

**Challenge of long texts.** The direct and superficial obstacle for long texts is that the pretrained *max position embedding* is usually 512 in BERT [12]. However, even if the embeddings for larger positions are provided, the memory consumption is unaffordable because all the activations are stored for back-propagation during training. For instance, a 1,500-token text needs about 14.6GB memory to run BERT-large even with batch size of 1, exceeding the capacity of common GPUs (e.g. 11GB for RTX 2080ti). Moreover, the $O(L^2)$ space complexity implies a fast increase with the text length $L$.

**Related works.** As mentioned in Figure 1, the *sliding window* method suffers from the lack of long-distance attention. Previous works [49, 33] tried to aggregate results from each window by mean-pooling, max-pooling, or an additional MLP or LSTM over them; but these methods are still weak at long-distance interaction and need $O(512^2 \cdot L/512) = O(512L)$ space, which in practice is still too large to train a BERT-large on a 2,500-token text on RTX 2080ti with batch size of 1. Besides, these *late-aggregation* methods mainly optimizes classification, while other tasks, e.g., *span extraction*, have $L$ BERT outputs, need O($L^2$) space for self-attention aggregation.

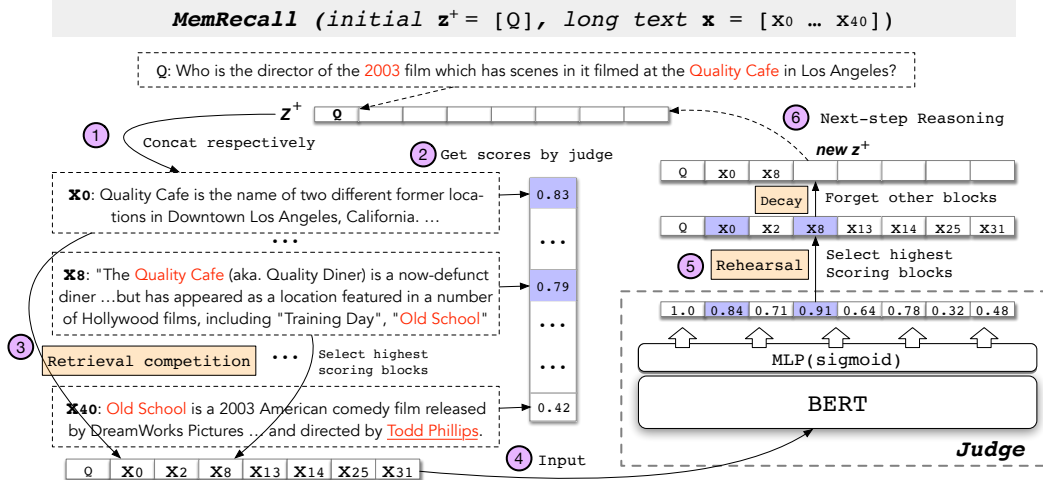

Figure 3: The MemRecall illustration for question answering. The long text $\mathbf{x}$ is broken into blocks $[\mathbf{x}_0 \ ... \ \mathbf{x}_{40}]$. In the first step, $\mathbf{x}_0$ and $\mathbf{x}_8$ are kept in $\mathbf{z}$ after rehearsal. The "Old School" in $\mathbf{x}_8$ will contribute to retrieve the answer block $\mathbf{x}_{40}$ in the next step. See Appendix for details.

In the line of researches to adapt transformers for long texts, many of them just compress or reuse the results of former steps and cannot be applied to BERT, e.g., Transformer-XL [8] and Compressive Transformer [37]. Reformer uses locality-sensitive hashing for content-based group attention, but it is not friendly to GPU and still needs verification for BERT usage. BlockBERT [35] cuts off unimportant attention heads to scale up BERT from 512 token to 1,024. The recent milestone longformer [4], customizes CUDA kernels to support window attention and global attention on special tokens. However, the efficacy of the latter are insufficiently investigated because the datasets are mostly in $4\times$ the window size of longformer. The direction of "lightweight BERTs" is promising but *orthogonal* to CogLTX, meaning that they can combine CogLTX to handle longer texts, so they will not be compared or discussed anymore in this paper. A detailed survey can be found in [25].

## 3 Method

### 3.1 The CogLTX methodology

This basic assumption of CogLTX is that "for most NLP tasks, a few key sentences in the text store *sufficient* and *necessary* information to fulfill the task". More specifically, we assume there exists a short text $\mathbf{z}$ composed by some sentences from the long text $\mathbf{x}$, satisfying

$$reasoner(\mathbf{x}^+) \approx reasoner(\mathbf{z}^+), \tag{1}$$

where $\mathbf{x}^+$ and $\mathbf{z}^+$ are inputs for the *reasoner* BERT w.r.t. the texts $\mathbf{x}$ and $\mathbf{z}$ as illustrated in Figure 2.

We split each long text $\mathbf{x}$ into blocks $[\mathbf{x}_0 ... \mathbf{x}_{T-1}]$ by dynamic programming (see the Appendix), which restricts the block length to a maximum of $B$, in our implementation $B = 63$ if the BERT length limit $L = 512$. The key short text $\mathbf{z}$ should be composed by some blocks in $\mathbf{x}$, i.e. $\mathbf{z} = [\mathbf{x}_{z_0} ... \mathbf{x}_{z_{n-1}}]$, satisfying $len(\mathbf{z}^+) \leq L$ and $z_0 < ... < z_{n-1}$. We denote $\mathbf{x}_{z_i}$ by $\mathbf{z}_i$. All blocks in $\mathbf{z}$ are automatically sorted to maintain the original relative ordering in $\mathbf{x}$.

The key-blocks assumption is strongly related to *latent variable models*, which are usually solved by EM [11] or variational bayes [19]. However, these methods estimate the distribution of $\mathbf{z}$ and require multiple-sampling, thus not efficient enough for BERTs. We take the essence of them into the design of CogLTX, and discuss the connections in § 3.3.

Two ingredients are essential in CogLTX, MemRecall and the joint training of two BERTs. As demonstrated in Figure 2, MemRecall is the algorithm utilizing the *judge* model to retrieve key blocks, which are fed into the *reasoner* to accomplish the task during inference.

### 3.2 MemRecall

As the brain recalls past episodes relevant to current information in working memory, the MemRecall aims to extract key blocks $\mathbf{z}$ from the long text $\mathbf{x}$ (see Figure 3).

**Input.** Although the aim is to extract key blocks, specific settings differ in the three types of tasks. In Figure 2 (a) (c), the question $Q$ or sub-sequence $\mathbf{x}[i]$ serves as query to retrieve relevant blocks. However, queries are absent in (b), and the relevance is only implicitly defined by the training data. For instance, sentences containing "Donald Trump" or "basketball" are more relevant for news topic classification than time reporting sentences. So how to seamlessly unify the cases?

MemRecall answers by accepting an initial $\mathbf{z}^+$ as an additional input besides $\mathbf{x}$. $\mathbf{z}^+$ is the short "key text" maintained during MemRecall to simulate working memory. The query in tasks (a)(c) becomes the initial information in $\mathbf{z}^+$ to provoke recalling. Then a *judge* model learns to predict task-specific relevance with the help of $\mathbf{z}^+$.

**Model.** The only model used by MemRecall is the *judge* mentioned above, a BERT to score the relevance for each token. Suppose $\mathbf{z}^+ = [\,\texttt{[CLS]}\ Q\ \texttt{[SEP]}\,\mathbf{z}_0\,\texttt{[SEP]}\dots\mathbf{z}_{n-1}]$,

$$judge(\mathbf{z}^+) = sigmoid(MLP(BERT(\mathbf{z}^+))) \in (0,1)^{len(\mathbf{z}^+)}. \tag{2}$$

The score of a block $\mathbf{z}_i$, denoted as $judge(\mathbf{z}^+)[\mathbf{z}_i]$, is the average of the scores of tokens in the block.

**Procedure.** MemRecall begins with a *retrieval competition*. Each block $\mathbf{x}_i$ is assigned a *coarse* relevance score $judge([\mathbf{z}^+\,\texttt{[SEP]}\,\mathbf{x}_i])[\mathbf{x}_i]$. The "winner" blocks with the highest scores are inserted into $\mathbf{z}$ as much as $len(\mathbf{z}^+) \leq L$. The superiority over *vector space models* [40] lies in that $\mathbf{x}_i$ fully interacts with current $\mathbf{z}^+$ via transformers, avoiding information loss during embedding.

The following *rehearsal-decay* period assigns each $\mathbf{z}_i$ a *fine* relevance score $judge(\mathbf{z}^+)[\mathbf{z}_i]$. Only the highest scored blocks are then kept in $\mathbf{z}^+$, just like the rehearsal-decay phenomenon in working memory. The motivation of fine scores is that the relative sizes of coarse scores are not accurate enough without interaction and comparison between blocks, similar to the motivation of reranking [7].

MemRecall in nature enables *multi-step reasoning* by repeating the procedure with new $\mathbf{z}^+$. The importance of iterative retrieval is highlighted by CogQA [13], as the answer sentence fails to be directly retrieved by the question in multi-hop reading comprehension. It is worth noting that blocks reserved from last step can also *decay*, if they are proved not relevant enough (with low scores) by more information from new blocks in $\mathbf{z}^+$, which is neglected by previous multi-step reasoning methods [13, 1, 10].

### 3.3 Training

The diversity of downstreaming tasks pose challenges for training (finetuning) the BERTs in CogLTX. The solutions under different settings are summarized in Algorithm 1.

**Supervised training for *judge*.** The span extraction tasks (Figure 2(a)) in nature suggest the answer block as *relevant*. Even multi-hop datasets, e.g. HotpotQA [53], usually annotate supporting sentences. In these cases, the *judge* is naturally trained in a supervised way:

$$loss_{judge}(\mathbf{z}) = CrossEntropy\big(judge(\mathbf{z}^+), relv\_label(\mathbf{z}^+)\big), \tag{3}$$

$$relv\_label(\mathbf{z}^+) = [\underbrace{1,\ 1,\ \dots,\ 1,}_{\text{for query}}\quad \underbrace{0,\ 0,\ \dots,\ 0,}_{\mathbf{z}_0\ is\ irrelevant}\quad \underbrace{1,\ 1,\ \dots,\ 1,}_{\mathbf{z}_1\ is\ relevant}\quad \dots] \in [0,1]^{len(\mathbf{z}^+)}, \tag{4}$$

where the training sample $\mathbf{z}$ is either a sequence of continous blocks $\mathbf{z}_{rand}$ sampled from $\mathbf{x}$ (corresponding to the data distribution of retrieval competition), or a mixture of all relevant and randomly selected irrelevant blocks $\mathbf{z}_{relv}$ (approximating the data distribution of rehearsal).

**Supervised training for *reasoner*.** The challenge for *reasoner* is to keep the consistency of data distributions during training and inference, which is a cardinal principle of supervised learning. Ideally, the inputs of *reasoner* should also be generated by MemRecall during training, but not all relevant blocks are guaranteed to be retrieved. For instance in question answering, if the answer block

---
**Algorithm 1:** The Training Algorithm of CogLTX
---
**Input:** Traing set $\mathcal{D} = [(\mathbf{x}_0, y_0), ..., (\mathbf{x}_n, y_n)]$, Hyperparameters: num_epoch, mode, $t_{up}, t_{down}$.

1 **if** *mode is unsupervised* **then**
2     Initialize the relevance labels in $\mathcal{D}$ by *Bm25* or *Glove* if possible. // see Appendix for details.
3 **for** *epoch from 1 to num_epoch* **do**
4     **for** $\mathbf{x}, y$ *in* $\mathcal{D}$ **do**
5       Extract a short ($len \leq L$) span from $\mathbf{x}$ at random as $\mathbf{z}_{rand}$.
6       $loss_{rand} = CrossEntropy(judge(\mathbf{z}_{rand}^+), relv\_label(\mathbf{z}_{rand}^+))$.
7       Aggregate all *relevant* blocks and some randomly chosen *irrelevant* blocks as $\mathbf{z}_{relv}$ ($len \leq L$).
8       $loss_{relv} = CrossEntropy(judge(\mathbf{z}_{relv}^+), relv\_label(\mathbf{z}_{relv}^+))$.
9       Update *judge* by descending $\nabla_\phi(loss_{rand} + loss_{relv})$. // $\phi$ is the parameters of *judge*.
10     **for** $\mathbf{x}, y$ *in* $\mathcal{D}$ **do**
11       Aggregate all *relevant* blocks in $\mathbf{x}$ as $\mathbf{z}$.
12       **for** *irrelevant block* $\mathbf{x}_i$ *in* $\mathbf{x}$ **do**
13         $score[\mathbf{x}_i] = judge([\mathbf{z}\ \mathbf{x}_i]^+)[\mathbf{x}_i]$ // can be replaced by cached scores during training *judge*.
14       Fill $\mathbf{z}$ up to length $L$ with highest scoring blocks. // corresponding to the $\mathbf{z}$ from MemRecall.
15       $loss = CrossEntropy(reasoner(\mathbf{z}^+), y)$.
16       Update *reasoner* by descending $\nabla_\theta loss$. // $\theta$ is the parameters of *reasoner*.
17       **if** *mode is unsupervised and epoch > 1* **then**
18         **for** *block* $\mathbf{z}_i$ *in* $\mathbf{z}$ **do**
19           $loss_{-\mathbf{z}_i} = CrossEntropy(reasoner(\mathbf{z}_{-\mathbf{z}_i}^+), y)$. // gradient-free, much faster than Line 15.
20           **if** $loss_{-\mathbf{z}_i} - loss > t_{up}$ **then** Label $\mathbf{z}_i$ as *relevant*;
21           **if** $loss_{-\mathbf{z}_i} - loss < t_{down}$ **then** Label $\mathbf{z}_i$ as *irrelevant*;

---

is missed by MemRecall, the training cannot proceed. Finally, an approximation is made to send all relevant blocks and the "winner" blocks in the retrieval competition to train the *reasoner*.

**Unsupervised training for *judge*.** Unfortunately, many tasks (Figure 2 (b)(c)) do not provide relevance labels. Since CogLTX assumes all relevant blocks *necessary*, we infer the relevance labels by interventions: test whether a block is indispensable by removing it from $\mathbf{z}$.

Suppose that $\mathbf{z}$ is the "oracle relevant blocks", according to our assumption,

$$loss_{reasoner}(\mathbf{z}_{-\mathbf{z}_i}) - loss_{reasoner}(\mathbf{z}) > t, \ \forall \mathbf{z}_i \in \mathbf{z}, \quad \text{(necessity)} \tag{5}$$

$$loss_{reasoner}([\mathbf{z}\ \mathbf{x}_i]) - loss_{reasoner}(\mathbf{z}) \approx 0, \forall \mathbf{x}_i \notin \mathbf{z}, \quad \text{(sufficiency)} \tag{6}$$

where $\mathbf{z}_{-\mathbf{z}_i}$ is the result of removing $\mathbf{z}_i$ from $\mathbf{z}$, and $t$ is a threshold. After every iteration in training *reasoner*, we ablate each block in $\mathbf{z}$, adjust its relevance label according to the increase of loss. Insignificant increase reveals the block as irrelevant, which will probably not "win the retrieval competition" again to train the *reasoner* in the next epoch, because it will be labeled as irrelevant to train the *judge* in the next epoch. Then real relevant blocks might enter $\mathbf{z}$ next epoch and be detected. In practice, we split $t$ into $t_{up}$ and $t_{down}$, leaving a buffer zone to prevent frequent changes of labels. We exhibit an example of unsupervised training on the 20News text classification dataset in Figure 4.

**Connections to latent variable models.** Unsupervised CogLTX can be viewed as a generalization of (conditional) latent variable model $p(y|\mathbf{x}; \theta) \propto p(\mathbf{z}|\mathbf{x})p(y|\mathbf{z}; \theta)$. EM [11] infers the distribution of $\mathbf{z}$ as posterior $p(\mathbf{z}|y, \mathbf{x}; \theta)$ in E-step, while variational bayes methods [19, 39] use an estimation-friendly $q(\mathbf{z}|y, \mathbf{x})$. However, in CogLTX $\mathbf{z}$ has a discrete distribution with up to $C_n^m$ possible values, where $n, m$ are the number of blocks and the capacity of $\mathbf{z}$ respectively. In some cases, sampling for hundreds of times to train BERTs might be required [19], whose expensive time consumption force us turn to *point estimation* for $\mathbf{z}$,[2] e.g. our intervention-based method.

The intervention solution, maintains an $\mathbf{z}$ estimation for each $\mathbf{x}$, and is essentially a local search specific to CogLTX. $\mathbf{z}$ is optimized by comparing nearby values(results after replacing irrelevant blocks) rather than Bayesian rules. The *judge* fits an inductive discriminative model to help infer $\mathbf{z}$.

| | Ep1 | Ep2 | Ep3 | | | Ep1 | Ep2 | Ep3 |
|---|---|---|---|---|---|---|---|---|
| | Score | Highest scoring blocks by judge | | | Score | Marked as irrelevant | Score | Marked as relevant |

The ground truth label: soc.religion.christian

| | | Ep1 | Ep2 | Ep3 | | | Ep1 | Ep2 | Ep3 |
|---|---|---|---|---|---|---|---|---|---|
| (1) | Harrassed at work, could use some prayers =CSE Dept., U.C. San… | | 0.16 | 0.19 | (7) | That is, someone that is supportive, comforting, etc. … healing… | 0.01 | | 0.09 |
| (2) | Yesterday I counted and realized that on seven different occasions… | | 0.16 | 0.16 | (8) | No one could be bothered to call me at the other building, even … | 0.01 | 0.13 | |
| (3) | If he/she does not seem to take any action, keep going up higher .. | | 0.12 | 0.14 | (9) | People in offices tend to be more insensitive while working than … | 0.01 | 0.12 | 0.08 |
| (4) | If you feel you can not discuss this with your boss, perhaps your … | | | 0.13 | (10) | Moderator allows me this latest indulgence. Well, if you can't turn … | 0.01 | | |
| (5) | It is unclear from your letter if you have done this or not. It is not … | 0.01 | | 0.13 | (11) | Then they will come back and wonder why I didn't want to go … | 0.01 | 0.14 | |
| (6) | If the company indeed does seem to want to ignore the entire… | 0.01 | | | (12) | They are doing it because they are still the playground bully … | | | |
| | | | | | (13) | In MY day, we had to make do with 5 bytes of swap… | | | 0.13 |

Figure 4: An example about unsupervised training of CogLTX on 20News dataset. All blocks are initialized as "irrelevant" by BM25 (no common words with the label *soc.religion.christian*). In the first epoch, the judge is nearly untrained and selects some blocks at random. Among them, (7) contributes most to the correct classification, thus is marked "relevant". In the second epoch, trained judge finds (1) with strong evidence "prayers" and (1) is marked as "relevant" at once. Then in the next epoch, (7) becomes not essential for classification and is marked as "irrelevant".

# 4   Experiments

We conducted experiments on four long-text datasets with different tasks. The token-wise (Figure 2 (c)) tasks are not included because they mostly barely need information from adjacent sentences, and are finally transformed into multiple sequence-level samples. The boxplot in Figure 5 illustrates the statistics of the text length in the datasets.

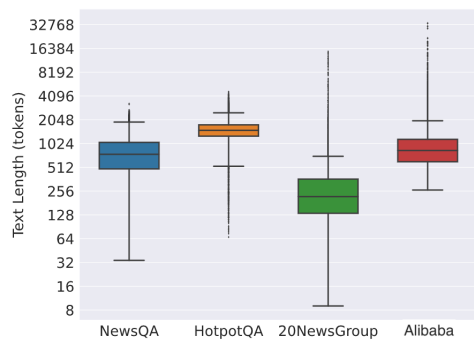

Figure 5: The boxplot of the text length distribution in the datasets.

In all experiments, the *judge* and *reasoner* are fine-tuned by Adam [18] with learning rate $4 \times 10^{-5}$ and $10^{-4}$ respectively. The learning rates warmup over the first 10% steps, and then linearly decay to $1/10$ of the max learning rates. The common hyperparameters are batch size $= 32$, *strides*$= [3, 5]$, $t_{up} = 0.2$ and $t_{down} = -0.05$.

In this section, we separately introduce each task with related results, analysis and ablation studies.

## 4.1   Reading comprehension

**Dataset and settings.**   Given a question and a paragraph, the task is to predict the answer span in the paragraph. We evaluate the performance of CogLTX on NewsQA [44], which contains 119,633 human-generated questions posed on 12,744 long news articles.[3] Since previous SOTA [43] is not BERT based (due to long texts) in NewsQA, to keep the similar scale of parameters for fair comparison, we finetune the *base* version of RoBERTa [26] for 4 epochs in CogLTX.

**Results.**   Table 1 show that CogLTX-base outperforms well-established QA models, for example BiDAF [41] (+17.8% $F_1$), previous SOTA DECAPROP [43], which incorporates elaborate self-attention and RNN mechanisms (+4.8%$F_1$), and even RoBERTa-large with sliding window (+4.8%$F_1$). We hypothesize that the first sentence (the lead) and the last sentence (the conclusion) are usually the most informative parts in news articles. CogLTX can aggregate them for reasoning while sliding window cannot.

Table 1: NewsQA results (%).

| Model | EM | $F_1$ |
|---|---|---|
| Match-LSTM [48] | 34.9 | 50.0 |
| BiDAF [41] | 37.1 | 52.3 |
| FastQAExt [51] | 42.8 | 56.1 |
| AMANDA [21] | 48.4 | 63.7 |
| MINIMAL [28] | 50.1 | 63.2 |
| DECAPROP [43] | 53.1 | 66.3 |
| RoBERTa-large [26] (sliding window) | 49.6 | 66.3 |
| CogLTX | **55.2** | **70.1** |

6

## 4.2 Multi-hop question answering

**Dataset and settings.** In complex scenarios, the answer is based on multiple paragraphs. Previous methods usually leverage the graph structure between key entities across the paragraphs [13, 36]. However, if we can handle long texts with CogLTX, the problem can be elegantly solved by concatenating all the paragraphs as the input of BERTs.

HotpotQA [53] is a multi-hop QA dataset of 112,779 questions, whose *distractor* setting provides 2 necessary paragraphs and 8 distractor paragraphs for each question. Both answers and *supporting facts* are required for evaluation. We treat each sentence as a block in CogLTX, and directly output the 2 blocks with the highest fine scores as supporting facts.

Table 2: Results on HotpotQA distractor (dev). (+hyperlink) means usage of extra hyperlink data in Wikipedia. Models beginning with "−" are ablation studies without the corresponding design.

| Model | Ans EM | Ans $F_1$ | Sup EM | Sup $F_1$ | Joint EM | Joint $F_1$ |
|---|---|---|---|---|---|---|
| Baseline [53] | 45.60 | 59.02 | 20.32 | 64.49 | 10.83 | 40.16 |
| DecompRC [29] | 55.20 | 69.63 | N/A | N/A | N/A | N/A |
| QFE [30] | 53.86 | 68.06 | 57.75 | 84.49 | 34.63 | 59.61 |
| DFGN [36] | 56.31 | 69.69 | 51.50 | 81.62 | 33.62 | 59.82 |
| SAE [45] | 60.36 | 73.58 | 56.93 | 84.63 | 38.81 | 64.96 |
| SAE-large | 66.92 | 79.62 | 61.53 | 86.86 | 45.36 | **71.45** |
| HGN [14] (+hyperlink) | 66.07 | 79.36 | 60.33 | 87.33 | 43.57 | 71.03 |
| HGN-large (+hyperlink) | 69.22 | 82.19 | 62.76 | 88.47 | 47.11 | **74.21** |
| *BERT (sliding window) variants* | | | | | | |
| BERT Plus | 55.84 | 69.76 | 42.88 | 80.74 | 27.13 | 58.23 |
| LQR-net + BERT | 57.20 | 70.66 | 50.20 | 82.42 | 31.18 | 59.99 |
| GRN + BERT | 55.12 | 68.98 | 52.55 | 84.06 | 32.88 | 60.31 |
| EPS + BERT | 60.13 | 73.31 | 52.55 | 83.20 | 35.40 | 63.41 |
| LQR-net 2 + BERT | 60.20 | 73.78 | 56.21 | 84.09 | 36.56 | 63.68 |
| P-BERT | 61.18 | 74.16 | 51.38 | 82.76 | 35.42 | 63.79 |
| EPS + BERT(large) | 63.29 | 76.36 | 58.25 | 85.60 | 41.39 | 67.92 |
| CogLTX | 65.09 | 78.72 | 56.15 | 85.78 | 39.12 | 69.21 |
| − multi-step reasoning | 62.00 | 75.39 | 51.74 | 83.10 | 35.85 | 65.35 |
| − rehearsal & decay | 61.44 | 74.99 | 7.74 | 47.37 | 5.36 | 37.74 |
| − train-test matching | 63.20 | 77.21 | 52.57 | 84.21 | 36.11 | 66.90 |

**Results.** Table 2 shows that CogLTX outperforms most of previous methods and all **7** BERT variants solutions on the leaderboard.[4] These solutions basically follow the framework of aggregating the results from sliding windows by extra neural networks, leading to bounded performances attributed to insufficient interaction across paragraphs.

The SOTA model HGN [14] leverages extra hyperlink data in Wikipedia, based on which the dataset is constructed. The thought of SAE [45] is similar to CogLTX but less general. It scores paragraphs by an attention layer over BERTs, selects the highest scoring 2 paragraphs and feeds them into BERT together. The supporting facts are determined by another elaborate graph attention model. With the well-directed designs, SAE fits HotpotQA better than CogLTX (2.2% Joint $F_1$), but does not solve the memory problem for longer paragraphs. CogLTX directly solves the multi-hop QA problem as ordinary QA, gets SOTA-comparable results and explains the supporting facts without extra efforts.

**Ablation studies.** We also summarize the ablation studies in Table 2, indicating that **(1)** multi-step reasoning does work (+3.9% Joint $F_1$) but not essential, probably because many questions themselves in HotpotQA are relevant enough with the second-hop sentences to retrieve them. **(2)** The metrics on supporting facts drop dramatically (-35.7% Sup $F_1$) without rehearsal for fine scores, because the relevance scores of top sentences are not comparable without attending to each other.

**(3)** As mentioned in § 3.3, the discrepancy of data distribution during training and test impairs the performance (-2.3% Joint $F_1$) if *reasoner* is trained by randomly selected blocks.

## 4.3 Text classification

**Dataset and settings.** As one of the most general tasks in NLP, text classification is essential to analyze the topic, sentiment, intent, etc. We conduct experiments on the classic 20NewsGroups [22], which contains 18,846 documents from 20 classes. We finetune RoBERTa for 6 epochs in CogLTX.

**Results.** Table 3 demonstrates that CogLTX, whose relevance labels are initialized by Glove [34], outperforms the other baselines, including previous attempts to aggregate the `[CLS]` pooling results from the sliding window [33]. Moreover, MLP or LSTM based aggregation cannot be trained end-to-end either on long texts.

**Ablation studies. (1)** Since the text lengths in 20 NewsGroups vary greatly (see Figure 5), we further test the performance only on the texts longer than 512 tokens (15%), which is even above the global result. **(2)** The initialization based on Glove provides good relevance labels, but the lack of adjustments by interventions still leads to 2.2% decrease in accuracy. **(3)** The Bm25 initialization is based on common words,

Table 3: 20NewsGroups results (%).

| Model | Accuracy |
|---|---|
| BoW + SVM | 63.0 |
| Bi-LSTM | 73.2 |
| fastText [16] | 79.4 |
| MS-CNN [32] | 86.1 |
| Text GCN [54] | 86.3 |
| MLP over BERT [33] | 85.5 |
| LSTM over BERT [33] | 84.7 |
| CogLTX (Glove init) | **87.0** |
| only long texts | 87.4 |
| − intervention (Glove init) | 84.8 |
| Bm25 init | 86.1 |

which only initializes 14.2% training samples due to the short label names, e.g., *sports.baseball*. The relevant sentences are inferred by interventions and the gradually trained *reasoner*, achieving an accuracy of 86.1%.

## 4.4 Multi-label classification

**Dataset and settings.** In many practical problems, each text can belong to multiple classes at the same time. The multi-label classification is usually transformed into binary classification by training an individual classifier for each label. Owing to the large capacity of BERT, we share the model for all the labels by prepending the label name at the beginning of the documents as input, i.e., `[[CLS] label [SEP] doc]`, for binary classification. Alibaba is a dataset of 30,000 articles extracted from an industry scenario in a large e-commerce platform. Each article advertises for several items from 67 categories. The detection of mentioned categories are perfectly modeled as multi-label classification. To accelerate the experiment, we respectively sampled 80,000 and 20,000 label-article pairs for training and testing. For this task, we finetune RoBERTa for 10 epochs in CogLTX.

**Results.** Table 4 shows that CogLTX outperforms common strong baselines. The word embeddings used by TextCNN [17] and Bi-LSTM are from RoBERTa for fair comparison. Even CogLTX-tiny (7.5M parameters) outperforms TextCNN. However, the max-pooling results of RoBERTa-large sliding window are worse than CogLTX (7.3% Macro-$F_1$). We hypothesize this is due to the tendency to assign higher probabilities to very long texts in max-pooling, highlighting the efficacy of CogLTX.

Table 4: Alibaba result (%).

| Model | Accuracy | Micro-$F_1$ | Macro-$F_1$ |
|---|---|---|---|
| BoW+SVM | 89.9 | 85.8 | 55.3 |
| Bi-LSTM | 70.7 | 62.1 | 48.2 |
| TextCNN | 95.3 | 94.1 | 91.3 |
| sliding window | 94.5 | 92.7 | 89.9 |
| CogLTX(tiny) | 95.5 | 94.4 | 92.4 |
| CogLTX(large) | **98.2** | **97.8** | **97.2** |

## 4.5 Memory and time consumption

**Memory.** The memory consumption of CogLTX is *constant* during training, superior to the $O(L^2)$ complexity of vanilla BERT. We also compare longformer [4], whose space complexity is roughly $O(L)$ if the number of global attention tokens is small relative to $L$ and independent of $L$. The detailed comparison is summarized in Figure 6 (Left).

**Time.** To accelerate the training of *reasoner*, we can cache the scores of blocks during training *judge* and then each epoch only needs $2\times$ time of single-BERT training. As the numbers of epochs until convergence are similar for CogLTX and sliding window in training, the main concern of CogLTX is the speed of inference. Figure 6 (Right) shows the time to process a 100,000-sample synthetic dataset with different text lengths. CogLTX, with $O(n)$ time complexity, is faster than vanilla BERT after $L > 2,048$ and approaches the speed of sliding window as the text length $L$ grows.

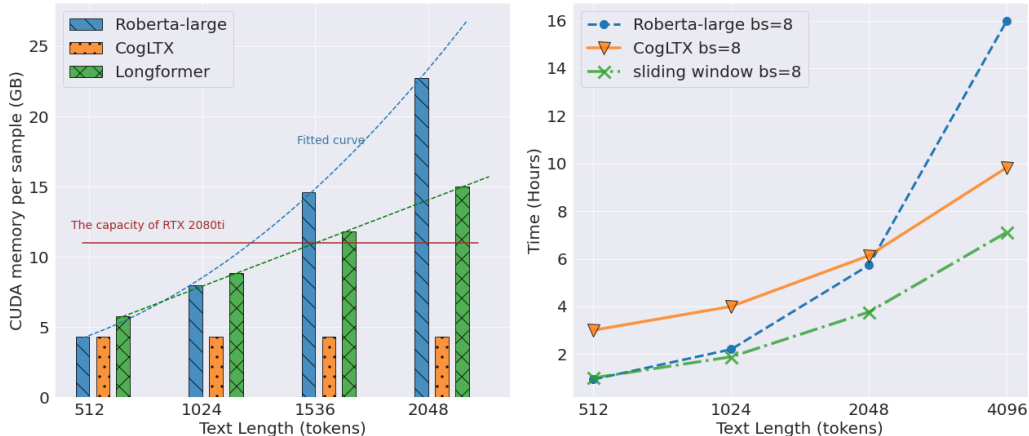

Figure 6: Memory and Time consumption with varying text length. The data about memory are measured with batch size = 1 on a Tesla V100. The batch size is 8 in the measurement of inference time consumption, and CogLTX does 1-step reasoning.

## 5   Conclusion and discussion

We present CogLTX, a cognition inspired framework to apply BERT to long texts. CogLTX only needs fixed memory during training and enables attentions between faraway sentences. Similar ideas were investigated on document-level in DrQA [6] and ORQA [23], and there are also previous works to extract important sentences in unsupervised ways, e.g. based on the metadata about structure [24]. Experiments on 4 different large datasets show its competitive performance. CogLTX is expected to become a general and strong baseline in many complex NLP tasks.

CogLTX defines a pipeline for long text understanding under the "key sentences" assumption. Extremely hard sequence-level tasks might violate it, thus efficient variational bayes methods (estimating the distribution of **z**) with affordable computation still worth investigations. Besides, CogLTX has a drawback to miss antecedents right before the blocks, which is alleviated by prepending the entity name to each sentence in our HotpotQA experiments, and could be solved by position-aware retrieval competition or coreference resolution in the future.

## Acknowledgements

The work is supported by NSFC for Distinguished Young Scholar (61825602), NSFC (61836013), and a research fund supported by Alibaba. The authors would like to thank Danqi Chen and Zhilin Yang for their insightful discussion, and responsible reviewers of NeurIPS 2020 for their valuable suggestions.

## Broader Impact

**Positive impact.** The proposed method for understanding longer texts is inspired by the theory of working memory in the human brain. After the success of pretraining language models that learn from extremely large corpus, it still remains mysterious how human being can memorize, understand, and conduct efficient yet effective reasoning process within a small memory budget, given a very few examples. Exploring such methods in fact may help design more elegant mechanism, or architecture that connects sub-models to solve complex tasks that require rich context and information. From a societal perspective, the proposed method can be also applied to many applications, e.g., legal document analysis, public opinion monitoring and searching.

**Negative impact.** With the help of such methods, social platforms may get better understanding about their users by analysing their daily posts. Longer texts understanding specifically provide more accurate and coherent interpretation of who they are, which is a privacy threat.

## Footnotes

[1]Codes are available at `https://github.com/Sleepychord/CogLTX`.

[2]analogous to K-means, which can be seen as EM for Gaussian mixture model with infinitesimal variances. Then the posterior of $\mathbf{z}$, mixture belonging, degenerates into the nearest cluster (the deterministic MLE value).

[3]We use the original version instead of the simplified version in MRQA [15], which removed long texts.

[4]https://hotpotqa.github.io

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
