[Supplementary Material]

# 6 Appendix

## 6.1 MemRecall

As described in § 3, the MemRecall is the process to extract the key blocks. We also need "strides" as input to indicate how many new blocks will be kept in each step. Details are showed as follows:

---

**Algorithm 2:** MemRecall

---

**Input:** Long text $\mathbf{x} = [\mathbf{x}_0 \dots \mathbf{x}_{n-1}]$, $strides = [stride_0, \dots, stride_{m-1}]$. // strides for m-step reasoning.

1   $\mathbf{z} = []$ // Initialize key blocks sequence as empty.
2   **for** *i from 0 to $m-1$* **do**
3     **for** *block $\mathbf{x}_i$ in $\mathbf{x}$* **do**
4       $score[\mathbf{x}_i] = judge([\mathbf{z} \ \mathbf{x}_i]^+)[\mathbf{x}_i]$ // relevance scores between $\mathbf{x}_i$ and query based on current $\mathbf{z}$.
5     Fill $\mathbf{z}$ up to length $L$ with highest scoring blocks. // Recall without interaction between candidate blocks.
6     $score[\mathbf{z}_0, \mathbf{z}_1 \dots] = judge(\mathbf{z}^+)$ // relevance scores after cross-attention and comparison.
7     Retain $\sum_{j=1}^{i} stride_j$ highest scoring blocks in $\mathbf{z}$. // rehearsal & decay, keep $stride_i$ more blocks.
8   **Return $\mathbf{z}$**.

---

## 6.2 Block Split

We predefine a cost for each punctuation and a basic cost for "hard truncation". The dynamic programming algorithm are showed as follows:

---

**Algorithm 3:** Block Split

---

**Input:** Long text $\mathbf{x}$, Punctuation costs *cost*, basic cost $c$, max block size $B$

1   Initialize $f[0] \dots f[B-1]$ as 0.
2   Initialize $from[0] \dots from[B-1]$ as $-1$.
3   **for** *i from $B$ to $len(\mathbf{x}) - 1$* **do**
4     $f[i] = +\infty$.
5     **for** *j from $i - B$ to $i - 1$* **do**
6       **if** *word is punctuation* **then**   $v = cost[word] + f[j]$ ;
7       **else** $v = c + f[j]$;
8       **if** $v < f[i]$ **then**
9         $f[i] = v, from[i] = j$.

10   $t = len(\mathbf{x}) - 1, blocks = []$.
11   **while** $t \geq 0$ **do**
12     prepend $\mathbf{x}[from[t] + 1 \dots t]$ to blocks.
13     $t = from[t]$.
14   **Return** $blocks$.

---

## 6.3 Initialization

In unsupervised training of CogLTX, an elaborate rather than random initialization for relevance labels could accelerate the convergence by large. Both query and textual label (e.g. label names for classification) can be used for initialization.

BM25 is a famous TF-IDF-like information retrieval method. Each block is scored based on the common words with query or textual label. However, the semantic relevance are neglected. For example, BM25 fails to find the relevance between label name "sports" with "baseball player".

Glove is a group of pretrained word representation. Suppose the query or textual label is $q$, and we score the relevance of a block $b$ by averaging the $len(q) \times len(b)$ inner products of their words. Top two blocks are initialized as relevant.