[Reviews · NeurIPS 2020]

Review 1

Summary and Contributions: This paper addresses an issue arising from the well-known quadratic space complexity of the attention mechanism. Attention is everywhere in modern sequence models such as Transformers, so that it has become difficult to handle longer sequences. The authors propose an elegant retrieval-like method that enables to integrate iterative "pondering" in scoring the possible subsequences. This relies on the assumption that within a longer text there exists a subsequence which is shorter than the length limit of pretrained Transformers and which is necessary and sufficient for performing a target task. Their method works as follows: - A sequence Z is initialized with the question (in the case of QA) or is initially empty (text classification). - All the subsequences are assigned coarse scores using a fine-tuned BERT judge, by concatenating them with Z. Only as many high-scoring subsequences are kept as they fit in a window. - All the high-scoring subsequences are embedded jointly and given a fine-grained score. The top k are kept and used to initialize Z again. - Loop until I iterations are done. - The final Z is provided to the BERT reasoner to perform extractive QA or text classification. The most enjoyable bit in the paper is the very clever trick to provide relevance labels to train the BERT judge in the case where the support sentences are not explicitly annotated (as in most QA datasets and in text classification), which exploits the difference between the BERT reasoner loss including and excluding a given subsequence.

Strengths: Quadratic space complexity is traded for linear time complexity, which is nice. The method is well-thought and results in nice boosts on the datasets considered in the evaluation setting. The trick used for the unsupervised training of the judge is worth at least the rest of the paper. It is also very appreciable to see that the authors have been inspired by the well-known principles in cognitive science, so that CogLXT is also theoretically motivated.

Weaknesses: 1) The assumption that within a longer text there exists a subsequence shorter than the length limit of pretrained Transformers that is necessary and sufficient for performing a target task is quite heavy. This holds more or less for QA, but will it generalize to long document summarization, for example? 2) The time complexity is identical to the sliding window approach, factoring out the number of iterative steps. However, there is quite a bit of overhead due to the iterative MemRecall mechanism. This overhead is only evaluated in Figure 5 with a batch size of 1, if I understand correctly. If so, I would have appreciated the results with a more realistic number of samples per batch. 3) The Longformer should have been used in the experiments as well as RoBERTa, as it has a space complexity of O(n log n). I am not convinced by the claim of the author(s) that this is completely orthogonal to their contribution. If comparable figures could be reached, this would weaken their empirical contribution. 4) The Reformer paper is referenced once in the introduction, without elaborating on it. I don't think the authors would be expected to compare against it, since no pre-trained checkpoint has been released, but I would have liked a discussion about it. 5) No qualitative assessment of the MemRecall mechanism is provided. It would have been nice to see what the unsupervised training of the BERT judge is selecting, especially for text classification, where you do not have support sentences (in which the answer occurs) -- this could be interesting also for interpretability, a topic which is completely unexplored.

Correctness: Yes, as far as I can tell.

Clarity: No. It took me quite a bit to figure out what the authors are doing, as many things are very hastily described, especially in the method exposition. The figures, though they could be certainly made clearer, help quite a bit, and are essential to the understanding. This should not be the case. Moreover, there are a number of typos/grammatical mistakes. The past participle of break is 'broken', not 'breaked'. 'RoBERTa' is always spelled 'Robert'/'Roberta'.

Relation to Prior Work: Yes with some caveats mentioned in 3) and 4) in the Weakness section.

Reproducibility: No

Additional Feedback: I would also explore what the approach offers for interpretability.


Review 2

Summary and Contributions: This paper considers applying BERT to long texts, by finding key related sentences for a target sentence. That is, among many sentences in a document, it extracts relevant context for a given sentence, and use them as context to limit the size of input and the memory consumption.

Strengths: - The approach is simple, and sound. - Finding the right context is an important problem, and this approach systematically solves it, in a way that can be applied to diverse corpora by dividing them into three types of tasks. - The evaluation is thorough, and it shows significant improvement in many tasks.

Weaknesses: - Training judge without labels can be expensive due to trial-and-error search of relevant sentences. - The sufficient condition Eq (6) is not explained. - Some more qualitative analysis in the evaluation can be added. For example, more example outputs of judge in the experiments for each task can be useful understanding the behavior. Also, accuracy breakdown based on document length (e.g., histogram) can be useful.

Correctness: Yes

Clarity: Yes

Relation to Prior Work: There are missing prior work considering long distance context. For example, "Unsupervised Sentence Embedding Using Document Structure-based Context" (ECML 2019) leverages document structure as a cue to find context. I wonder if the proposed approach actually extracts something similar to that can be extracted by document structure information.

Reproducibility: Yes

Additional Feedback: ------------After rebuttal--------------- I appreciate the authors' detailed response to many of the concerns. Assuming the authors reflect these and also other comments in the reviews to the camera ready version, I would still accept this paper. ----------------------------------------------- - Please see the weakness section above. - Initially, z+ passed to judge contains all sentences. How do you avoid the memory problem with judge? - "We hypothesize that the first sentence (the lead) and the last sentence (the conclusion) are usually the most informative parts in news articles." Can a document structure based approach, or even "first and last sentences" baseline does similarly?


Review 3

Summary and Contributions: The paper proposes a model that given a long text and its blocks/sentences and their relevance judgements (which can be initialized from IR methods) iteratively refines the relevance scores and find supporting sentences for a downstream task. The model uses two BERT models, a "judge" for scoring, and "reasoner" for, e.g., answering. The main contribution is the fine-tuning/training iterative procedure and good results on a number of datasets and tasks.

Strengths: The model is well evaluated, including ablation studies for the model features. The iterative application of retrieval/reasoning using pretrained(BERT) models seems to be novel.

Weaknesses: Some parts of important related work are missing: Please discuss the relation of your model to "Latent Retrieval for Weakly Supervised Open Domain Question Answering" (ACL 2019), which is a single step paragraph retrieval and reasoning and was applied to a different set of tasks, while this work has multiple iterations of similar retrieval/reasoning and considers sentences and is applied to different datasets (application to multi-hop datasets is nice), but former would have been good as a benchmark.

Correctness: Looks good.

Clarity: The paper is well written.

Relation to Prior Work: The paper discusses previous work but could discuss it in more detail and could really benefit from discussion of recent open domain QA work (e.g. one mentioned above), sentence extraction / summarization, and also more detailed discussion of recent methods scaling attention to long sequence (some of which are cited, e.g. Reformer).

Reproducibility: Yes

Additional Feedback: - Please add more details to the paper on the relevance scores; are they binary? How exactly are they updated?


Review 4

Summary and Contributions: ################## After rebuttal ############################### Thank the authors for their response. Concerns about baselines that authors can consider to improve the paper. For Table 3 (20NewsGroups), it is better for the authors to offer another version of CogLTX with BERT for fair comparison with the baselines since RoBERTa significantly outperforms BERT on GLUE leaderboard (88.1 vs 80.5). For Table 4 (A+), why sliding window uses BERT but CogLTX uses RoBERTa. It is better for sliding window to uses RoBERta too. On the other hand, for table 2 (HotpotQA), I am convinced by the comparison between Longformer and CogLTX on HotpotQA (69.5 vs 69.2) (discussion in the rebuttal). Thanks for the authors' response. I improve the overall score from 5 to 6. ######################################################### The paper introduces a method called CogLTX to apply BERT/RoBERTa to long texts. Experiments show the proposed method outperforms BERT sliding window methods but underperforms SOTA models.

Strengths: 1. Figure 3 is really clear which helps me understand the main idea of the paper. 2. Experiments results on different tasks are shown. 3. The paper proposes unsupervised training for the judge process to solve the tasks without relevance labels.

Weaknesses: 1. The main idea is similar to SAE just as the authors say in the paper. SAE scores paragraphs and CogLTX cores sentences. Compared with SAE, CogLTX is a more complex and fine-grained method, but the performance is worse than that of SAE. It is better to analyze memory, computation, and inference latency between SAE and CogLTX. 2. Although the paper shows experimental results of 4 tasks. However, except HotpotQA task, the baselines of the other three tasks are not strong with a poor backbone (like [29] in Table 3 uses BERT, but CogLTX uses RoBERTa that is much better than BERT) or just sliding window (Table 1 and Table 4). It is better to clarify the backbone of baseline and CogLTX. As for Hotpot QA task (Table 2), SAE can be seen as a fair baseline to CogLTX, but the performance of CogLTX is worse than that of SAE and still has a not small margin (69.21 vs 71.45).

Correctness: Yes

Clarity: Yes, I like the Figure 3.

Relation to Prior Work: Yes

Reproducibility: Yes

Additional Feedback:

[Author Response · NeurIPS 2020]

Thank you very much for your careful, insightful and valuable comments, we will explain your concerns point by point.

**Common questions: 1. Qualitative Examples.** (a) A hard unsupervised training case. (b) Updated Figure 5.

| Score | Highest scoring blocks by judge | Score | Marked as irrelevant | Score | Marked as relevant |
|---|---|---|---|---|---|

(a) An unsupervised example in 20news class "soc.religion.christian"

| | | Ep1 | Ep2 | Ep3 | | | Ep1 | Ep2 | Ep3 |
|---|---|---|---|---|---|---|---|---|---|
| (1) | Harrassed at work, could use some prayers =CSE Dept., U.C. San... | | 0.16 | 0.19 | (7) | That is, someone that is supportive, comforting, etc. ... healing... | 0.01 | | 0.09 |
| (2) | Yesterday I counted and realized that on seven different occasions... | | 0.16 | 0.16 | (8) | No one could be bothered to call me at the other building, even ... | 0.01 | 0.13 | |
| (3) | If he/she does not seem to take any action, keep going up higher .. | | 0.12 | 0.14 | (9) | People in offices tend to be more insensitive while working than ... | 0.01 | 0.12 | 0.08 |
| (4) | If you feel you can not discuss this with your boss, perhaps your ... | | | 0.13 | (10) | Moderator allows me this latest indulgence. Well, if you can't turn ... | 0.01 | | |
| (5) | It is unclear from your letter if you have done this or not. It is not ... | 0.01 | | 0.13 | (11) | Then they will come back and wonder why I didn't want to go ... | 0.01 | 0.14 | |
| (6) | If the company indeed does seem to want to ignore the entire... | 0.01 | | | (12) | They are doing it because they are still the playground bully ... | | | |
| | | | | | (13) | In MY day, we had to make do with 5 bytes of swap... | | | 0.13 |

All blocks are initialized as "irrelevant" by BM25 (no common words with the label "soc.religion.christian"). In the 1st epoch, the judge is nearly untrained and selects some blocks at random. Among them, (7) contributes most to the correct classification, thus is marked "relevant". In the 2nd epoch, trained judge finds (1) with strong evidence "prayers" and (1) is marked as "relevant" at once. Then in the next epoch, (7) becomes not essential for classification and is marked as "irrelevant".

**2. Detailed discussion about related works. #1**: Reformer uses LSH for content-based grouping attention, but it is not
friendly to GPU, weakens position relevance and may tend to be finetuned to a local-minimum grouping. It still needs
verification for BERTs. Longformer mixes global and window attention. It ($O(L \log L)$ space) is contemporaneous with
CogLTX ($O(1)$ space) but ArXived in advance. It performs similar to CogLTX on HotpotQA (69.5 vs 69.2). Its window
size is 512 and most HotpotQA samples < 2,048, so whether faraway sentences in longer texts can fully interact really
via global attention is still unknown, but CogLTX **can seamlessly combine it ("Orthogonal")** to handle longer texts.
**#2**: The structure-based contexts (ECML'19) is indeed relevant, but it relies on Metadata and 3 manual cases, and
CogLTX is more general. We will discuss it in the camera-ready version. **#3**: Yes, ORQA did partly inspire this work,
but it focuses on retrieval via BERT embeddings (fast), while CogLTX is for long texts in reader period (fine-grained).
**To Reviewer#1: 1. About the assumption.** We agree there are some tasks violating the assumption, but to the best
of our knowledge, the assumption can be applied to **most** of the common NLP tasks (paper Figure 2) for long texts,
including summarization. One of the most popular summarization setting is "extractive summarization"[1], aiming to
select important sentences to form a summarization. "Abstractive summarization" also mainly used key sentences[2].
[1] Text summarization with pretrained encoders. [2] Bottom-up abstractive summarization. EMNLP'18.
**2. Time consumption for batch size > 1.** See Figure (b). The time of batch size = 8 shows a similar trend as = 1 (the
same total number of samples). We also compare the space of Longformer, which is still much heavier than CogLTX.
**3. No enough details to fully reproduce.** The main concerns are about details for unsupervised mode. We will
definitely add details and polish the writing in the camera-ready version. The codes will be open-sourced too.
**To Reviewer#2: 1. Expensive trial-and-error search without labels.** Not so much. The trials only need "model
inference" and are gradient-free, which is much faster than training with data-flow graph (Algo 1 Line 19). In
experiments, it only cost $\sim 2\times$ time of training with labels, instead of $N\times$ time (N is the number of blocks).
**2. Explain sufficient condition in Eq(6).** This means some key sentences **z** are *enough* for the task, more sentences
are useless(won't reduce the loss). See Figure (a) for case study.
**3. Memory concerns during initally judging $z^+$.** This is in the "model inference" period, when memory issue is not
so serious(in training, sentences are separated with their relevance labels as different samples). We can also split them
into different batches in the *retrieval competition* step to keep fixed memory overhead.
**To Reviewer#3: 1. Explain details about relevant scores.** Yes, they are binary and updated by *intervention*, a.k.a.
removing it from **z** and to see the change of loss (Line 139 and Algo1 Line 18-21), which is fast (Reviewer#2 **1.**).
**To Reviewer#4:** CogLTX is a **general** framework to apply BERTs to arbitrarily long texts **without memory concerns**
and retain the long-distance attention.
**1. Comparison with SAE. (1)** CogLTX is more **general** than SAE, which is specific to HotpotQA. It **cannot** be used
for other tasks, e.g. classification, let alone **unsupervised** cases. SAE selects top 2 paragraphs because the HotpotQA is
constructed by 2 of 10 paragraphs. It uses an extra GCN on graphs built with 3 kinds of co-occurrence of entities from
spaCy NER package, with a module for Yes/No over the GCN. These designs are hard to be applied to other datasets.
Different from SAE, CogLTX concatenates the paragraphs as a normal document into a universal pipeline without extra
tricks. **(2)** SAE **does not** completely solve the memory problem. Actually, SAE-large needs V100 or better GPUs to
train HotpotQA, and could raise OOM for longer or more paragraphs. See answer **3.** for further discussion.
**2. Weak BERT backbone for baselines on 3 tasks.** This might be a misunderstanding. We did use RoBERTa for all
baselines (described in Line 184, Line 253), except the baseline in Task 3, whose results are from the original paper.
**3. Discussion on Model over BERT.** Some concerns might origin from the comparison with "Model over BERT"
baselines, cutting documents into segments and aggregating BERT results by another model. They **don't** really solve
the memory problem, but **sacrificing early interactions** (Line 69)(7 such methods worse than CogLTX in HotpotQA).
End2end training needs $O(512^2 \cdot L/512) = O(512L)$ space. It usually only improves the max length $2\times \sim 4\times$(depend on
device and model size) for batch size = 1 and less for batch size > 1. As an advantage, CogLTX and sliding window only
need **constant space**, especially fit for real-world data. Besides, "Model over BERT" mainly optimizes classification.
Other tasks, like *span extraction*, has $L$ BERT outputs, still need $O(L^2)$ space for self-attention aggregation.

[Meta-Review · NeurIPS 2020]

This paper proposes a method for dealing with the quadratic complexity of attention with the goal of extending Transformers to longer sequences. The authors proposes a clever way to train a 'judge' for selecting relevant subsequences. The idea is novel and seems to work well in practice. Please make sure to address all suggestions by the reviewers in the final version of the paper.